# Cargo Recognition Mechanisms of Yeast Myo2 Revealed by AlphaFold2-Powered Protein Complex Prediction

**DOI:** 10.3390/biom12081032

**Published:** 2022-07-26

**Authors:** Yong Liu, Lingxuan Li, Cong Yu, Fuxing Zeng, Fengfeng Niu, Zhiyi Wei

**Affiliations:** 1SUSTech-HIT Joint PhD Program, Harbin Institute of Technology, Harbin 150001, China; 11849501@mail.sustech.edu.cn; 2Department of Biology, School of Life Sciences, Southern University of Science and Technology, Shenzhen 518055, China; 12032126@mail.sustech.edu.cn (L.L.); yuc@sustech.edu.cn (C.Y.); zengfx@sustech.edu.cn (F.Z.); 3Brain Research Center, School of Life Sciences, Southern University of Science and Technology, Shenzhen 518055, China; 4Guangdong Provincial Key Laboratory of Cell Microenvironment and Disease Research, Southern University of Science and Technology, Shenzhen 518055, China; 5Shenzhen Key Laboratory of Cell Microenvironment, Southern University of Science and Technology, Shenzhen 518055, China

**Keywords:** Myo2p, organelle transport, molecular motor, cytoskeleton, globular tail domain (GTD), protein–protein interaction, protein structure prediction

## Abstract

Myo2, a yeast class V myosin, transports a broad range of organelles and plays important roles in various cellular processes, including cell division in budding yeast. Despite the fact that several structures of Myo2/cargo adaptor complexes have been determined, the understanding of the versatile cargo-binding modes of Myo2 is still very limited, given the large number of cargo adaptors identified for Myo2. Here, we used ColabFold, an AlphaFold2-powered and easy-to-use tool, to predict the complex structures of Myo2-GTD and its several cargo adaptors. After benchmarking the prediction strategy with three Myo2/cargo adaptor complexes that have been determined previously, we successfully predicted the atomic structures of Myo2-GTD in complex with another three cargo adaptors, Vac17, Kar9 and Pea2, which were confirmed by our biochemical characterizations. By systematically comparing the interaction details of the six complexes of Myo2 and its cargo adaptors, we summarized the cargo-binding modes on the three conserved sites of Myo2-GTD, providing an overall picture of the versatile cargo-recognition mechanisms of Myo2. In addition, our study demonstrates an efficient and effective solution to study protein–protein interactions in the future via the AlphaFold2-powered prediction.

## 1. Introduction

Intracellular trafficking is required for various cellular activities, in which cytoskeletal motors, including F-actin-based myosins and microtubule-based kinesins and dyneins, recognize and transport numerous materials to proper locations in cells [1,2]. A central question in motor-based transport is how the motor proteins specifically recognize their cargos. Over the past two decades, extensive structural studies have found certain motor/cargo interactions to partially address this question [3,4,5]. However, the highly diverse motor/cargo interactions and the difficulties in protein sample preparation and/or structure determination limit the comprehensive understanding of the cargo-recognition mechanisms of cytoskeletal motors.

Class V myosins are well-characterized for cargo recognition, due to their remarkable capability in transporting a diverse array of cellular cargos, such as organelles, vesicles, protein complex and mRNAs [6]. In *Saccharomyces cerevisiae*, Myo2, a class V myosin, has been found to transport mitochondria, secretory vesicles, vacuoles, trans-Golgi membranes, peroxisomes, and spindles [7,8]. Many cargo adaptors in yeast, including mitochondrial Myo2 receptor-related 1 (Mmr1) [9], inheritance of peroxisomes gene 2 (Inp2) [10], suppressor of myosin (Smy1) [11], vacuole-related protein 17 (Vac17) [12], Karyogamy protein 9 (Kar9) [13], the polarisome subunit Pea2 [14], and several Rab proteins (e.g., Sec4, Ypt32 and Ypt11) [15,16,17], have been identified as the binding partners of Myo2 for its cargo recognition, which makes Myo2 an ideal target in studying motor/cargo interactions. 

The very C-terminal globular tail domain (GTD) is the major cargo-binding region in Myo2 [18], which contains subdomain-I and subdomain-II (Figure 1A). Biochemical and mutagenesis studies have indicated that Myo2 utilizes three different binding sites on the GTD to interact with different cargo adaptors [19] (Figure 1A). Furthermore, our previous crystallographic study of Myo2-GTD in complex with three cargo adaptors (Mmr1, Smy1 and Inp2) revealed the molecular basis of these motor/cargo interactions [20]. Nevertheless, as the identified cargo adaptors of Myo2 share little sequence similarity, the recognition mechanism of Myo2 remains unknown for most cargos. Obtaining the Myo2-GTD structure in complex with cargo adaptors is the crucial approach to uncover the cargo-binding modes of Myo2. 

Recently, AlphaFold2 and RoseTTAFold have been emerging as powerful tools in protein structure prediction with a high accuracy [21,22]. Based on these, applications including ColabFold [23] and AlphaFold-Multimer [24] were subsequently developed to predict the structures of protein complexes. These progresses in protein structure prediction provide a potential solution to quickly obtain the atomic structures of the Myo2-GTD/adaptor complexes. Therefore, we evaluated the validity of AlphaFold2-based prediction of the Myo2-GTD/adaptor complexes by using the crystal structures of Myo2-GTD in complex with Mmr1, Smy1, and Inp2 as benchmarks. With the optimized strategy of protein complex prediction, we uncovered the previously unknown binding modes of Vac17, Kar9 and Pea2 to Myo2-GTD, which were strongly supported by our biochemical analyses. Based on these results, we found that the reliable predictions highly depend on certain factors governing protein–protein interactions, such as binding affinity and stoichiometry. Finally, by integrally analyzing the binding of the six cargo adaptors to Myo2-GTD, we further explicitly showed the structural similarities and differences among the versatile Myo2/cargo interactions.

## 2. Materials and Methods

### 2.1. Complex Structure Prediction

Structure predictions were performed in the standalone platform of ColabFold [23], which was set up in a local computer with Linux operating system and accelerated with one NVIDIA GeForce RTX 2080 Ti GPU. In each complex structure prediction, the amino acid sequences of Myo2-GTD and a cargo adaptor with indicated boundary (Appendix A) were inputted into ColabFold with a binding ratio of 1:1, except for the complex prediction of Myo2-GTD and Pea2-CC1 dimer with a binding ratio of 1:2 and the recycle number (the iterative time that the predicted structure will be fed back for refinement) was set as 3, 6, 12 or 24, respectively. During the calculation, the monomeric neural network model was employed, as was the method of MMseqs2 (Many-against-Many sequence searching) [25] for searching against protein sequence databases including UniRef90 [26], and the value of pLDDT was used to rank the predicted models with different levels of confidence. Finally, five models with their histogram plots of PAE value, which shows the confidence level (blue to red indicates confident to unconfident) with regard to the relative position of two regions in the protein or protein complex, thus implying the potential contacts of these two regions, were output for each prediction condition (Appendix A).

### 2.2. Local RMSD and Interface CAD-Score Calculations 

To indicate the structural consensus of the cargo adaptors in the predicted models of the complexes, the local RMSD was calculated by superimposing the Myo2-GTD structures in the complex models based on their C_α_ atoms using the *align* method and then comparing the C_α_ position of the two corresponding fragments in the cargo adaptors (Appendix A). The local RMSD of the MISs of Mmr1, Smy1, and Inp2 were calculated between each predicted model and the corresponding crystal structures. The local RMSD of the MISs of Vac17, Kar9, and Pea2 and the per residue Cα atom distance of input sequences were calculated between each of the two predicted models. The RMSD were calculated using the *rms_cur* method and the per residue Cα atom distance were determined using *RmsdByResidue*. All the operations mentioned above were carried out in PyMOL v. 2.4.1 (https://pymol.org).

To compare the inter-chain contacts, the interface CAD-score (CAD-score^iface^) was calculated in an evaluation mode of inter chain interfaces between the crystal structures and the predicted models of Myo2-GTD and Mmr1/Smy1/Inp2 complex, or between each of the two predicted models of Myo2-GTD and the Vac17/Kar9/Pea2 complex, based on the online server (https://bioinformatics.lt/cad-score/, accessed on 16 July 2022) [27].

### 2.3. Plasmids

The genes of Myo2, Vac17, Kar9 and Pea2 were amplified from yeast genomic DNA (*Saccharomyces cerevisiae*, *strain S288c*) and then their indicated boundaries were cloned into a modified pET32a vector with a tandem Trx and His_6_ tags at the N-terminus. For GST-Pea2-CC1 expression, the Pea2-CC1 sequence was constructed into a vector of pGEX-4T-1 with an N-terminal GST tag. All the mutants were generated by QuickChange Site-Directed Mutagenesis kit and were verified by sequencing.

### 2.4. Protein Expression and Purification

All protein fragments were overexpressed with an induction of 0.2 mM isopropyl-β-d-thiogalactoside (IPTG) at 16 °C for 20 h in BL21(DE3) *E. coli* cells. Myo2, Vac17, Kar9, and Pea2 with indicated boundaries and mutations were purified by Ni^2+^-NTA affinity chromatography (GE Healthcare, Chicago, Illinois, USA), followed by size exclusion chromatography (Superdex^TM^ 200 pg, GE Healthcare) with a buffer of 50 mM Tris pH7.5, 100 mM NaCl, 1 mM DTT and 1 mM EDTA. GST-Pea2-CC1 was purified by Glutathione Sepharose 4 column (GE Healthcare), followed by size exclusion chromatography with the same buffer. The protein purities were checked by SDS-PAGE.

### 2.5. Isothermal Titration Calorimetry (ITC)

ITC measurements were performed on a PEAQ-ITC Microcal calorimeter (Malvern Panalytical, Malvern, UK). The protein samples of 200 μM Vac17-MIS and 500 μM Myo2-GTD were prepared in the syringe to titrate the proteins of 20 μM Myo2-GTD and 50 μM Kar9-MIS or Pea2-MIS in the cell at 25 °C, respectively. For each titration, the protein of 3 μL in the syringe was injected into the cell with an interval of 150 s between two injections for baseline recovery. The titration data were analyzed using a build-in ITC analysis software and fitted by a one-site binding model.

### 2.6. Analytical Size Exclusion Chromatography (aSEC)

aSEC was performed on an ÄKTA pure system (GE Healthcare) by loading 100 μL of indicated protein samples with a final concentration of 60 μM onto a Superdex 200 Increase 10/300 GL column (GE Healthcare), equilibrated with a buffer containing 50 mM Tris-HCl pH 7.5, 100 mM NaCl, 1 mM EDTA, and 1 mM DTT.

### 2.7. Multi-Angle Static Light Scattering 

The molecular weights of Trx-Myo2-GTD, Trx-Vas17-MIS and their mixture, as well as GST-tagged or Trx-tagged Pea2-CC1 were measured by the multi-angle static light scattering method, respectively. A total of 100 μL of protein samples with a final concentration of 100 μM were loaded onto the same aSEC system described above and the elution was analyzed by the tandem DAWN-HELEOS-II and Optilab T-rEX (Wyatt Technology Corporation, Santa Barbara, CA, USA) for molar mass measurement.

## 3. Results

### 3.1. Benchmark of AlphaFold2-Based Prediction with the Solved Complex Structures of Myo2-GTD

ColabFold is a fast (~20–30-fold faster than AlphaFold2) and easy-to-use tool used for predicting the 3D structures of both single-chain proteins and protein complexes, which was developed to allow efficient AlphaFold2-based predictions with limited computational resources [23]. With the input amino acid sequences of target proteins, ColabFold performs calculations guided by the settings of the binding ratio and iterative recycles. The ColabFold outputs include the predicted 3D models of target proteins or protein complexes, the value of the predicted Local Distance Difference Test (pLDDT), which indicates the prediction confidence for each residue in each model, and a matrix plot of Predicted Aligned Error (PAE), which suggests the potential contacts between each of the two residues [23] (Appendix A). By coloring the predicted model according to the pLDDT values, we can directly visualize the prediction confidence of different regions in the target protein, such as the successfully predicted Myo2-GTD structure (Figure 1A).

We tested the feasibility of using ColabFold to predict the structures of Myo2-GTD in complex with the cargo adaptors. Mmr1, Smy1, and Inp2 were chose for the trials, as their crystal structures in complex with Myo2-GTD that were determined previously can be used to evaluate the ColabFold prediction results. Considering the fact that our prediction strategy should be feasible for unsolved complex structures, instead of inputting the short sequences used for crystallization, we subjected the Myo2-interacting sequences (MISs) of Mmr1 [19], Smy1 [28], and Inp2 [10] that were mapped biochemically (Appendix A) into ColabFold together with the Myo2-GTD sequence. As increasing the iterative recycles may improve the prediction accuracy [23], the prediction of each Myo2-GTD/MIS complex was performed with different recycle numbers. For each prediction, the Myo2-GTD structures of the five output models were aligned for the following analysis (Appendix A).

In the Myo2-GTD/Mmr1-MIS complex prediction, the predicted interfaces between Myo2-GTD and Mmr1-MIS are highly similar to that in the crystal structure, which was indicated by the interface CAD-score of ~0.6 (see [27] for a detailed introduction of CAD-score) and almost all the interacting regions of Mmr1-MIS fragments show an essentially identical Myo2-GTD-bound conformation (Cα-RMSD of ~1 Å) to that observed in the crystal structure (Figure 1B,C and Appendix A). The prediction of five models can be reached to convergence with six or more iterative recycles. In the Myo2-GTD/Smy1-MIS complex prediction, three out of five predicted models resemble the Smy1-MIS conformation in the crystal structure (Figure 1B,D and Appendix A). To further evaluate the molecular details of the predicted interactions, we compared the interface residues between the predicted and crystal structures. As shown in Figure 1E,F, the sidechains of the predicted interface residues are well-aligned with those in the crystal structures, confirming that the bindings of Mmr1-MIS and Smy1-MIS to Myo2-GTD can be successfully predicted with accurate atomic details using ColabFold. In addition, the GTD-bound sequences of either Mmr1-MIS or Smy1-MIS in the correctly predicted models have the pLDDT and PAE values outstanding in the whole MIS sequence (Appendix A), suggesting that the pLDDT and PAE values may be useful in identifying the binding sequence in the complex structure prediction.

In contrast to the above two cases, the Myo2-GTD/Inp2-MIS complex prediction failed to reproduce the Inp2-MIS orientation observed in the crystal structure (Figure 1B and Appendix A), although most predicted models of Inp2-MIS are located near the Inp2-binding groove at site I on Myo2-GTD (Figure 1A and Appendix A) [20]. Two of the predicted Inp2-MIS fragments are positioned at site I in the reverse direction to that in the crystal structure (Appendix A), presumably due to the impact of the sidechain orientation of Y1287 in Myo2-GTD. To accommodate the bulky sidechain of a phenylalanine in Inp2-MIS, Y1287 rotates to another direction, as observed in the crystal structure of Inp2-bound Myo2-GTD, and such a rotation of Y1287 was also found in the crystal and predicted structures of Smy1-bound Myo2-GTD (Appendix A). However, in all predicted Myo2-GTD/Inp2-MIS models, Y1287 adopts the same orientation as the apo-GTD structure and clashes with the correct GTD-bound conformation of Inp2-MIS (Appendix A), and thereby may cause the wrong prediction. Additionally, compared with the successful predictions above, we found that the pLDDT scores of the predicted Inp2-MIS structures are relatively low and only two out of five predicted models share similar conformations (Appendix A), supporting the idea that the pLDDT score and the consensus of five models at the binding region are the indicators in validating prediction results (Appendix A).

### 3.2. Myo2-GTD/Vac17 Complex Prediction

Although Vac17 is a well-characterized cargo adaptor for Myo2 in vacuole transport via its binding to Myo2-GTD [12], the molecular basis underlying the specific recognition of Vac17 by Myo2-GTD remains largely unknown. Since Vac17-MIS has been narrowed down to the region containing residues 112-157 [19] and forms a stable complex with Myo2-GTD (Appendix A), we tried to determine the Myo2-GTD structure in complex with Vac17-MIS. However, extensive crystallization trials using different Vac17 fragments failed to yield any crystals. Thus, we applied the above strategy to predict the structure of the Myo2-GTD/Vac17-MIS complex by ColabFold. To evaluate the predictions, we calculated the interface CAD-scores between each of the two predicted complex models and the local RMSD values between the GTD-binding regions of each of the two predicted Vac17-MISs in these aligned structures (Appendix A).

As a result, all predicted models share an essentially identical conformation of Vac17-MIS after the six-recycle calculation, suggesting a successful prediction of the Myo2-GTD/Vac17-MIS complex’s structure (Figure 2A and Appendix A). In this predicted complex, Vac17-MIS contains an N-terminal loop and a C-terminal α-helix. The highly conserved region, consisting of the C-terminal part of the loop and the N-terminal part of the α-helix, binds to site II of Myo2-GTD (Figure 2A,B). Fully consistent with a previous mutagenesis study of Myo2-GTD [19], the predicted Vac17-binding surface on Myo2-GTD covers all the residues identified for Vac17 binding (Figure 2C). This surface contains a negatively charged patch composed of E1293^GTD^, D1296^GTD^ and D1297^GTD^, which complementarily interacts with the positively charged residues R142^Vac17^ and K138^Vac17^ (Figure 2D,E). Notably, D1297^GTD^ not only forms two hydrogen bonds (Figure 2E), but also attracts the positively charged N-terminus of the α-helix of Vac17-MIS (Figure 2D). This structural finding explains why the substitution of D1297^GTD^ to a similar asparagine can abolish the Myo2-GTD/Vac17 interaction [19]. In addition to the polar interactions, several conserved hydrophobic residues in Vac17-MIS pack into a hydrophobic groove on site II of Myo2-GTD (Figure 2B,F), which also creates a hydrophobic environment surrounding D1297^GTD^ to further strengthen the polar interaction between D1297^GTD^ and Vac17-MIS. 

To confirm our prediction of the Myo2-GTD/Vac17-MIS complex, we measured the binding affinities of Myo2-GTD to Vac17-MIS variants. The isothermal titration calorimetry (ITC)-based analysis showed a strong interaction between Myo2-GTD and Vac17-MIS (*K*_d_ of ~0.3 μM) (Figure 2G). Consistent with our structural observations, either the charge-reversing mutations (e.g., R135E, K138E, and R142E) or the hydrophobic-to-hydrophilic mutations (e.g., L137Q and I139Q) in Vac17-MIS eliminate the Myo2-GTD/Vac17-MIS interaction (Figure 2H). Hence, our computational, structural and biochemical analyses successfully reveal the unprecedented details of the Myo2-GTD/Vac17-MIS interaction. 

### 3.3. Myo2-GTD/Kar9 Complex Prediction: Prediction-Assistant Identification of the Binding Site 

Myo2 has been reported to regulate the spindle orientation during yeast mitosis through its binding to Kar9 [13]. The molecular basis underlying the Myo2/Kar9 interaction is still unclear. With our success in the prediction of the Myo2-GTD/Vac17-MIS complex, we next tried to predict the Myo2-GTD/Kar9-MIS complex. However, the MIS region in Kar9 has not been identified, despite the N-terminal folded domain (Kar9^N^) being suggested to play a role in the Myo2/Kar9 interaction [29]. Surprisingly, although the binding of Kar9^N^ to Myo2-GTD (*K*_d_ of ~16 μM) was supported by our ITC-based analysis, the unstructured C-terminal region of Kar9 (Kar9^C^) shows a stronger binding to Myo2-GTD (*K*_d_ of ~3 μM) (Figure 3A). Thus, we focused on the identification of the MIS region in Kar9^C^.

Consistent with our ITC data, Kar9^C^ was predicted to interact with Myo2-GTD as suggested by the PAE plot from the ColabFold output that uses either the full-length Kar9 or Kar9^C^ sequence as input (Appendix A). However, as the predicted binding sequence is not consistent among the five models, we designed more truncated fragments of Kar9^C^ for additional predictions (Figure 3A). As indicated by our predictions, a region containing residues 460–512 (Kar9^C^-MIS) that is conserved across different yeast species may be important for the binding of Myo2-GTD to Kar9 (Figure 3A,B and Appendix A). To confirm the prediction-based mapping results, we performed binding affinity measurements between the Kar9^C^ fragments and Myo2-GTD (Figure 3A and Appendix A). Indeed, the short Kar9^C^-MIS fragment binds to Myo2-GTD with a *K*_d_ value comparable to Kar9^C^ (Figure 3A). Notably, Kar9^C^-MIS has no sequence overlapped with the two reported regions of Kar9^C^ that bind to the microtubule-binding protein Bim1, an EB1 orthologue in yeast (Figure 3A) [30,31], supporting the previous finding that Kar9 can simultaneously interact with both Myo2 and Bim1 to coordinate the actin filament and microtubule in the cell cycle [13].

In all output models after the calculations with the iterative recycle number larger than 3, the Kar9^C^-MIS regions with high pLDDT scores are well aligned with each other, indicating the high reliability of the complex structure prediction (Figure 3C and Appendix A). In the complex structure, the N-terminal region of Kar9^C^-MIS folds as an α-helix plus a following short β-sheet and packs with the site I cleft of Myo2-GTD (Figure 3C,D). The site I cleft provides a largely hydrophobic environment to accommodate several conserved hydrophobic residues (e.g., L466, L470, L471, M474, and I476) in Kar9^C^-MIS (Figure 3B,E). Meanwhile, the polar interactions in the two ends of the α-helix of Kar9^C^-MIS, especially the salt bridge formed between K473^Kar9^ and E1272^Myo2^, orientate this α-helix in the site I cleft (Figure 3F). Additionally, the C-terminal sequence extending from the α-helix of Kar9^C^-MIS forms a short β-sheet and interacts with the loop-I/II connecting subdomains I and II (Figure 3F), further enhancing the binding specificity. 

Consistent with the complex structure, either the hydrophobic-to-hydrophilic mutations L466Q and L470Q or the charge-reversed mutation K473E in Kar9^C^-MIS abolish the interaction between Kar9^C^-MIS and Myo2-GTD (Figure 3G,H). To confirm if Kar9^C^-MIS binds to site I of Myo2-GTD, we measured the binding of Kar9^C^-MIS to Smy1-MIS::Myo2-GTD, which was designed by fusing the Smy1-MIS sequence to the N-terminus of Myo2-GTD [20]. As Smy1-MIS also binds to site I of Myo2-GTD, this fusion protein loses its capability to interact with the site-I-binding adaptors [20]. As expected, the fusion protein shows no detectable binding to Kar9^C^-MIS (Figure 3I). Together, we identified a previously unknown interaction between Myo2-GTD and Kar9^C^-MIS by combining the complex structure prediction and biochemical verification.

### 3.4. Myo2-GTD/Pea2 Complex Prediction: Determination of Binding Stoichiometry 

Pea2, a subunit of polarisome in yeast, was recently found to interact with Myo2, which focuses the polarisome at the tips of yeast cells and is required for proper budding [14]. Although the GTD in Myo2 and the C-terminal region in Pea2 has been identified for this interaction [14], their binding mode is still unknown. Our sequence analysis and structure prediction indicated that the mapped GTD-binding fragment of Pea2 is mainly comprised of a conserved coiled coil (CC1) containing residues 235–327, which may form a dimer (Figure 4A,B and Appendix A). Thus, we fed the Pea2-CC1 sequence into ColabFold and tried the binding ratio of both 1:2 and 1:1 for the structure prediction of Myo2-GTD in complex with the dimeric and monomeric Pea2-CC1, respectively. 

In each of the predictions with six or more iterative recycles, three predicted models of the dimeric Pea2-CC1 are docked at site III of Myo2-GTD with a highly similar orientation (Figure 4C and Appendix A). Importantly, the interface between Pea2-CC1 and Myo2-GTD in these models covers the Pea2-binding surface on Myo2-GTD that was mapped by mutagenesis [14] (Figure 4D). In contrast to the high consensus among the predicted models of Myo2-GTD in complex with the Pea2-CC1 dimer, the complex structure prediction with a 1:1 binding ratio failed to produce any consensus binding mode between the monomeric Pea2-CC1 and Myo2-GTD (Appendix A), suggesting the importance of Pea2-CC1 dimerization in the Myo2-GTD/Pea2-CC1 complex formation. Consistently, the interface analysis shows that both two protomers (Pea2-1 and Pea2-2) in the Pea2-CC1 dimer are involved in the binding of Pea2-CC1 to Myo2-GTD (Figure 4B,E,F). Several charged residues from both Pea2-1 and Pea2-2 form salt bridges with their counterparts in site III of Myo2-GTD (Figure 4E). Among these residues, R270^Pea2−2^ plays a critical role in the interaction by forming a salt bridge with E1484^Myo2^ and packing with W1407^Myo2^ through the cation–π interaction. In addition, R270^Pea2−2^ interacts with D269^Pea2−1^ to stabilize the Pea2-CC1 dimer. In agreement with the previous identification of the importance of several hydrophobic residues (L1331, F1334, L1411, and Y1415) on the surface of site III of Myo2-GTD [14], these residues form a hydrophobic surface patch to interact with several hydrophobic residues of Pea2-CC1, especially F261^Pea2−1^, which is closely surrounded by the hydrophobic residues on the surface of site III (Figure 4F). 

To test the reliability of the complex structure formed between Myo2-GTD and the Pea2-CC1 dimer, we purified Pea2-CC1 and measured its molecular weight by using size exclusion chromatography coupled with multi-angle light scattering. Unexpectedly, the result showed that Pea2-CC1 is a monomer (Appendix A), suggesting that Pea2-CC1 alone may not form a stable dimer in solution. The ITC-based binding analysis indicated a weak interaction (*K*_d_ of ~50 μM) between Myo2-GTD and the Pea2-CC1 monomer (Figure 4G). To enhance the potential dimer formation of Pea2-CC1, we added an N-terminal GST tag that has a weak dimerization propensity. The GST-tagged Pea2-CC1 forms a predominant dimer (Appendix A) and shows a 5-fold enhancement of the binding affinity to Myo2-GTD (Figure 4H), supporting the idea that the dimerization of Pea2-CC1 is crucial for the binding of Pea2 to Myo2-GTD. Consistent with our structural analysis, the R270E and F261Q mutations that presumably disrupt the hydrophilic and hydrophobic interactions between Myo2-GTD and Pea2-CC1, respectively, eliminate the binding of Pea2-CC1 dimer to Myo2-GTD, as indicated by the ITC analysis (Figure 4I). These results further confirm the importance of Pea2-CC1 dimerization for Myo2-GTD binding, as F261 and R270 are located at the GTD-binding sites of Pea2-1 and Pea2-2 protomers, respectively (Figure 4B). Correspondingly, the Myo2-GTD/Pea2-CC1 interaction was also blocked by the site III mutations L1331S and W1407A (Figure 4J). Interestingly, although the Pea2-CC1 dimer should have two identical GTD-binding surfaces, our ITC data showed that the binding ratio was 1:2 rather than 2:2, indicated by the N value of ~0.5 (Figure 4H). We suspected that the binding of GTD on one GTD-binding surface of the Pea2-CC1 dimer may slightly change the Pea2-CC1 conformation, which influences the other GTD-binding surface and inhibits the Pea2-CC1 dimer to bind to the second GTD. Indeed, we found that the conformational change between the apo and GTD-bound Pea2-CC1 structures (Figure 4K). Thus, our structural analysis reveals how Myo2 employs site III of the GTD to specifically recognize a cargo adaptor and suggests a convenient approach to determine the binding stoichiometry through the structure prediction in ColabFold.

### 3.5. Comparison of the Six Cargo-Binding Modes of Myo2-GTD

With the three predicted structures of Myo2-GTD in complex with Vac17, Kar9 and Pea2 and the three crystal structures of Myo2-GTD in complex with Mmr1, Smy1 and Inp2 [20], we were able to structurally compare the six cargo-binding modes of Myo2-GTD. As summarized in Figure 5A, Smy1, Inp2, and Kar9^C^ bind to site I of Myo2-GTD, Mmr1 and Vac17 bind to site II of Myo2-GTD, and Pea2 binds to site III of Myo2-GTD. These structural findings are fully consistent with our ITC-based analysis (Figure 5B and Appendix A), in which the GTD constructs having cargo-binding defects in sites I, II, and III, respectively, were applied.

At site I of Myo2-GTD, which is largely hydrophobic (Figure 3E), although the MIS fragments of Smy1, Inp2, and Kar9^C^ are completely overlapped with each other (Figure 5A), these MISs occupy the different surface areas on Myo2-GTD (Figure 5C). Kar9^C^-MIS occupies the largest surface area among these three binding modes, elucidating at least in part the reason that the GTD binding of Kar9^C^-MIS is much stronger than those of Smy1-MIS and Inp2-MIS (Figure 5B). Differing from the extended loop structures of Smy1-MIS and Inp2-MIS, Kar9^C^-MIS adopts the helical conformation, which allows more residues (17 residues in Kar9^C^-MIS while ~10 residues in Smy1-MIS and Inp2-MIS) to fit in the site I cleft and thereby increases the contact between Kar9^C^-MIS and Myo2-GTD (Figure 5A). Nevertheless, these three MISs show a similar distribution of four hydrophobic residues in the interfaces (circled in Figure 5D), indicating the core role of these residues in maintaining the GTD/MIS interactions at site I. In addition, Kar9^C^-MIS and Smy1-MIS have another overlapped hydrophobic residue for GTD binding and Kar9^C^-MIS possesses a unique hydrophobic residue for GTD binding (indicated by arrows in Figure 5D), further explaining the different binding strengths of the three cargo adaptors to site I of Myo2-GTD. 

At site II of Myo2-GTD, which contains the negatively charged and neutral surface patches on the right and left sides, respectively (Figure 2D and Figure 5E), the binding surfaces for Vac17-MIS and Mmr1-MIS are partially overlapped and both have a buried area of ~800 Å^2^ (Figure 5E). Interestingly, although Vac17-MIS and Mmr1-MIS adopt a similar fold having an N-terminal loop and a C-terminal α-helix upon binding to Myo2-GTD, they bind to Myo2-GTD in the reverse directions to each other (Figure 5A,F). The helices of Vac17-MIS and Mmr1-MIS are positioned on the right and left sides of site II, respectively, while the N-terminal loops are overlapped (Figure 5E,F). As the surface charge of site II is unevenly distributed, the helices of Mmr1-MIS and Vac17-MIS bind to site II in different modes. The helix in Vac17-MIS is positioned nearly perpendicularly to the site II surface and thereby poses its positively charged N-terminus to contact the negatively charged surface patch on the left side, whereas the helix in Mmr1-MIS is placed parallelly to the site II surface to allow more hydrophobic contacts on the right side (Figure 5E,F) [20]. In contrast to the different GTD-binding mode of the helices, the N-terminal loops of Vac17-MIS and Mmr1-MIS are similar in their binding to Myo2-GTD, as indicated by three overlapped interface residues (circled in Figure 5F), despite that the two loops have the completely reversed directions. Additionally, to further cover the both side of the site II, Mmr1-MIS and Vac17-MIS have R409^Mmr1^ and F132^Vac17^ in the loop N-termini (indicated by open arrows in Figure 5F) to interact with the left (negatively charged) and right (hydrophobic) sides of site II, respectively. 

In addition to Pea2, Kar9 and several Rabs have been reported to interact with Myo2-GTD via site III and the Kar9-binding surface on site III was reported to overlap with the Pea2-binding surface [19] (Figure 5G). However, our prediction trials failed to capture the interaction of Kar9 at site III (Appendix A). As Kar9^N^ also associates with Myo2-GTD (Figure 3A), we speculated that Kar9^N^ may bind to the site III. In line with our hypothesis, the site III mutations that interfere with the Myo2-GTD/Pea2-CC1 interaction also disrupt the binding of Myo2-GTD to Kar9^N^ (Figure 5B). Interestingly, Kar9^N^ adopts a helix bundle structure [29] (Figure 5H), which can be partially aligned with the coiled-coil structure of Pea2-CC1 (Figure 5H), implying that Kar9^N^ may bind to site III of Myo2-GTD through a mode similar to that of Pea2-CC1.

## 4. Discussion

Protein–protein interactions govern almost all cellular activities. Understanding the molecular basis of protein–protein interactions is essential for addressing fundamental questions in cell biology. However, due to various difficulties in obtaining homogenous protein complexes for structure determination (e.g., low protein yield, bad sample quality, and high cost for experimental resources and facility usages), only a very limited portion of protein–protein interactions have been structurally characterized. In this study, we successfully obtained several reliable complex structures by AlphaFold2-based predictions, all of which were finished within hours using an affordable computer. In addition to providing the binding details, the AlphaFold2-powered prediction facilitates the determination of the binding region and stoichiometry, which is also time consuming in bench work.

Whether the predicted model is reliable is a major issue in protein structure predictions. Based on the predictions performed in this study, we demonstrated that not only the consensus of predicted models but also the pLDDT value of the binding region and the PAE map are useful to address the reliability concern. In the benchmark predictions, we found that compared with the regions that are not involved in the binding to Myo2-GTD, the GTD-binding regions in the adaptors have high pLDDT scores and low values in the PAE map (Appendix A). In our reliable predictions, the GTD-binding regions are all highly conserved across different species (Figure 2B, Figure 3B and Figure 4B). Considering that the target-binding sequence may be conserved through evolution, the sequence conservation can be used to further evaluate the predicted interaction. 

In the complex prediction, ColabFold provides several variables (e.g., input sequence, binding ratio, and iterative cycle) for users to adjust, which can be optimized to obtain more reliable results (Appendix A). First, the input sequence length affects the successful rate of a prediction. In the prediction of the Myo2-GTD and Kar9 complex, we tested the Kar9 boundaries with different sequence lengths (Figure 3A). As indicated by the PAE maps from the boundaries containing the Kar9^C^-MIS sequence, the shorter boundaries have more models to show obvious contacts between Myo2-GTD and Kar9^C^-MIS (Appendix A). The disordered sequence of Kar9^C^ that is not involved in GTD binding may have a chance to nonspecifically pack with the MIS sequence in the structure prediction, thereby interfering with the prediction of intermolecular interactions. Second, the binding ratio is critical for the effective prediction of protein–protein interactions. As we showed in the prediction of the Myo2-GTD and Pea2-CC1 complex, Pea2-CC1 must form a dimer to create the proper surface for the recognition of Myo2-GTD (Figure 4C). Thus, for a complex prediction with unknown binding stoichiometry, it would be worthwhile to try a different binding ratio. Third, the iterative calculation with sufficient recycles in the prediction improves the prediction. In our prediction cases, the default three-recycle calculation sometimes failed to generate the consensus models, whereas increasing the recycle number to six or more results in a better prediction in most cases.

Apart from the adjustable inputs, the intrinsic property of protein–protein interactions, such as binding affinity, also influences the complex prediction. The MIS fragments of Mmr1, Smy1, and Inp2 show the different GTD-binding affinities from high to low (Figure 5B). In the benchmark prediction, Mmr1-MIS and Smy1-MIS adopt the correct GTD-binding conformation in most models after the six-recycle calculation, while no model shows a correct GTD-binding conformation of Inp2-MIS even with a 24-recycle calculation (Figure 1B). Similarly, all five models reach consensus after a six-recycle calculation for Vac17-MIS and Kar9^C^-MIS with the relatively high GTD-binding affinities (Figure 5B). Hence, in the AlphaFold2-based complex prediction, it is difficult to accurately predict weak associations (*K*_d_ > 20 μM) and the prediction of a weak interaction should be carefully validated. 

By comparing the Myo2-GTD structures in complex with six cargo adaptors, we systematically analyzed the different cargo recognition mechanisms of Myo2. As the cargo binding concentrates on the three sites of Myo2-GTD (Figure 5A), some cargo adaptors (e.g., Smy1/Inp2/Kar9 at site I or Mmr1/Vac17 at site II) may compete with each other when loaded onto Myo2. Indeed, the competition between Mmr1 and Vac17 for Myo2 regulates the volume of mitochondria and vacuoles in yeast budding [19]. A similar cargo competition was observed between MICAL1 and Spires in binding to the GTD of Myosin Va, which is required for Rab11-coated vesicle loading and unloading [32]. Conversely, cargo adaptors bind to Myo2-GTD via different sites may be involved in cooperative cargo transport. For example, Smy1 binding to Myo2-GTD has been reported to enhance the association of Myo2 and Sec4 (a yeast Rab protein) on the vesicle surface [11]. Hence, the six structures summarized here provide important information that will aid in understanding the relationship between the large number of Myo2-transported cargos. Additionally, Kar9 has two distinct sequences that interact with sites I and III of Myo2-GTD (Figure 5B). As Inp2 was also found to interact with Myo2-GTD via both sites I and III [20,33], it is likely that certain cargo adaptors use two binding sites to enhance the cargo-recognition specificity. 

In addition, Myo2-GTD in the tail has been considered as a brake to inhibit Myo2 activity via the direct interaction with Myo2-Motor in the head and two key residues, K1473 and R1402, on Myo2-GTD have also been identified [34]. Interestingly, the head/tail binding surface on Myo2-GTD is closed to site III, which suggests that the cargo recruitments (e.g., Pea2) via site III on Myo2-GTD would potentially release the head/tail interaction to activate Myo2 for cargo transport. 

## Figures and Tables

**Figure 1 biomolecules-12-01032-f001:**
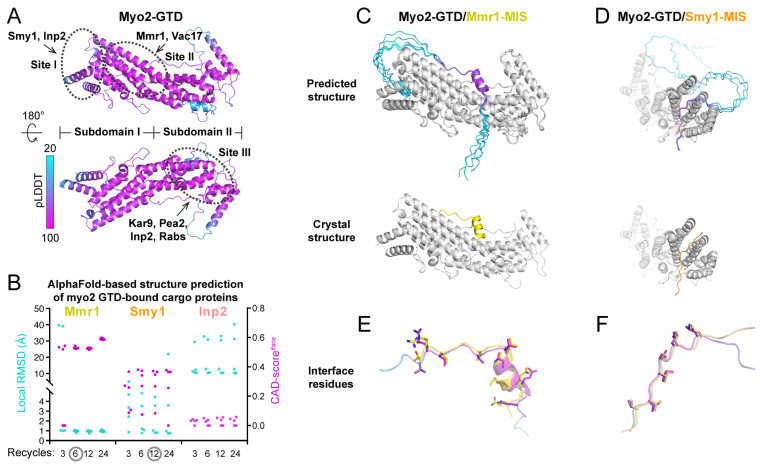
Benchmarks of the structure prediction of Myo2-GTD in complex with three cargo adaptors. (**A**) The predicted Myo2-GTD structure. The structure is colored according to the value of pLDDT and the three reported cargo-binding sites were indicated by dashed circles. (**B**) Summary of the interface CAD-scores (CAD-scores^iface^) calculated between the crystal structures and the predicted models and the local RMSD values calculated between the GTD-binding regions of Mmr1, Smy1 and Inp2 in the crystal structures of Myo2-GTD in complex with these adaptors (PDB IDs: 6IXP, 6IXQ and 6IXR, respectively) and the corresponding regions of the five predicted complex models from each prediction condition with the calculation of 3, 6, 12 and 24 iterative recycles, respectively. (**C**,**D**) The structural comparisons between the predicted and crystal structures of Myo2-GTD in complex with Mmr1-MIS (**C**) and Smy1-MIS (**D**). The predicted complex models showing low local RMSD values were superimposed by aligning their Myo2-GTD structures. The MIS fragments in the predicted models are colored according to their pLDDT values. (**E**,**F**) The comparison of the interface residues of Mmr1-MIS (**E**) or Smy1-MIS (**F**) between the predicted model and the crystal structure.

**Figure 2 biomolecules-12-01032-f002:**
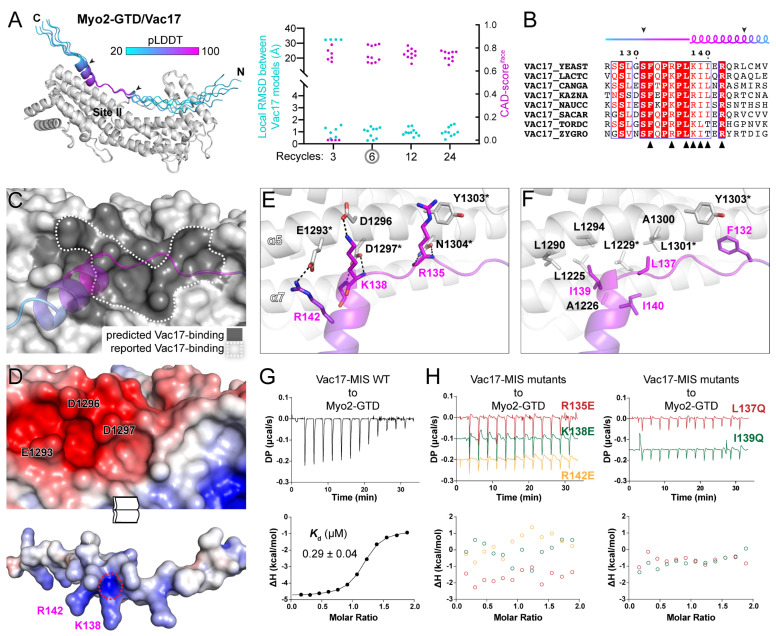
Prediction and analysis of the Myo2-GTD and Vac17-MIS complex. (**A**) The five predicted structures of Myo2-GTD and Vac17-MIS complex. The structures were superimposed by aligning their Myo2-GTD structures and Vac17-MIS is colored according to pLDDT values of residues. The GTD-binding boundary of Vac17 is indicated by two arrowheads and was used for local RMSD calculation between each of the two predicted models. The CAD-scores^iface^ were calculated between each of the two predicted models. (**B**) Sequence alignment of Vac17-MIS among different species. The secondary structure with GTD-binding boundary indicated by two arrowheads were depicted and the interface residues, as shown in (**E**,**F**), were labeled by black triangles. (**C**) The Vac17-binding surface on Myo2-GTD. The predicted Vac17-binding surface covers the reported one. (**D**) Open-book view of the electrostatic potential surfaces of Myo2-GTD and Vac17-MIS. The key interface residues for charge–charge interaction are labeled and the positively charged N-terminus of the helix in Vac17-MIS is circled by a red dash line. (**E**,**F**) Molecular details of the Myo2-GTD/Vac17-MIS interface in the predicted structure, including the polar (**E**) and hydrophobic (**F**) interactions. The reported interface residues are marked with asterisks. (**G**,**H**) ITC-based measurements of the binding affinities between wild-type (**G**) or mutated (**H**) Vac17-MIS and Myo2-GTD.

**Figure 3 biomolecules-12-01032-f003:**
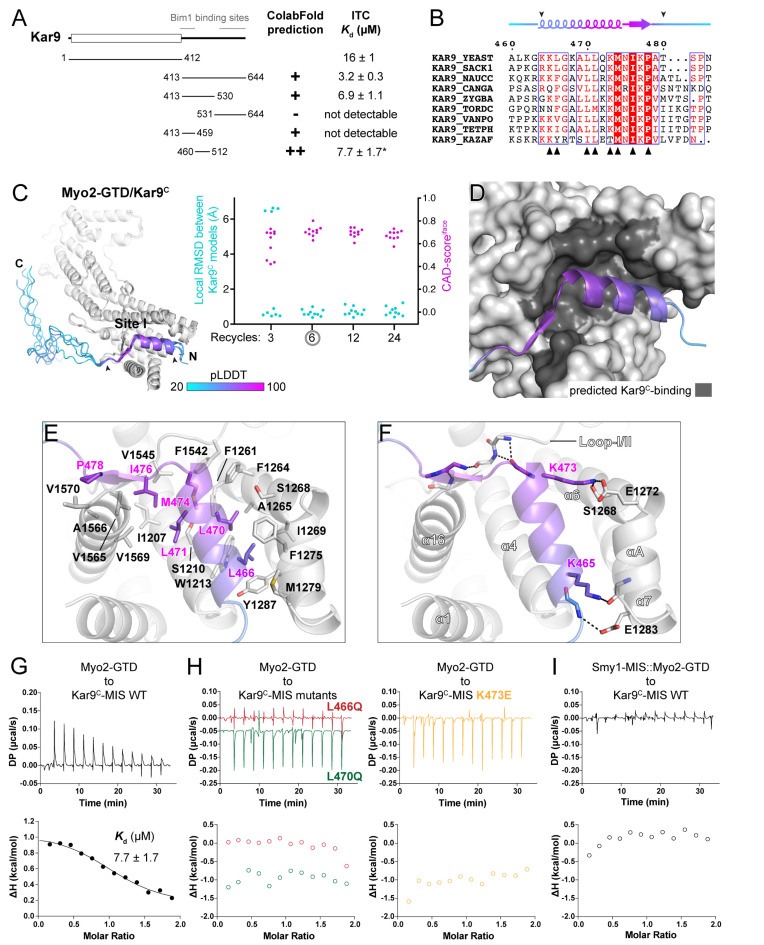
Prediction and analysis of the Myo2-GTD and Kar9^C^-MIS complex. (**A**) The identification of the GTD-binding region in Kar9. The GTD-binding potentials of different Kar9 boundaries were investigated by protein complex prediction in ColabFold and then verified by ITC-based affinity measurement. Two Bim1 binding sites in Kar9 are also indicated, which show no overlap with Kar9^C^-MIS. (**B**) Sequence alignment of Kar9^C^-MIS among different species. The interface residues, as shown in (**D**), were indicated by black triangles. The GTD-binding boundary was indicated by two arrowheads. (**C**) The five predicted structures of Myo2-GTD and Kar9^C^-MIS complex. The structures were superimposed by aligning their Myo2-GTD structures and Kar9^C^-MIS is colored according to pLDDT values of residues. The GTD-binding boundary of Kar9^C^ was indicated by two arrowheads and was used for local RMSD calculation. The CAD-scores^iface^ were calculated between each of the two predicted models. (**D**) The Kar9^C^-binding surface on Myo2-GTD. (**E**,**F**) Molecular details of the Myo2-GTD/Kar9^C^-MIS interface in the predicted structure, including the hydrophobic (**E**) and polar (**F**) interactions. The loop I/II connecting subdomains I and II in Myo2-GTD is indicated. (**G**,**H**) ITC-based measurements of the binding affinity between wild-type (**G**) or mutated (**H**) Kar9^C^-MIS and Myo2-GTD. (**I**) The disruptive binding between Kar9^C^-MIS and the Smy1-MIS::Myo2-GTD fusion protein, which was designed by fusing the Smy1-MIS sequence to the Myo2-GTD N-terminus to prevent site I of Myo2-GTD from binding to other proteins.

**Figure 4 biomolecules-12-01032-f004:**
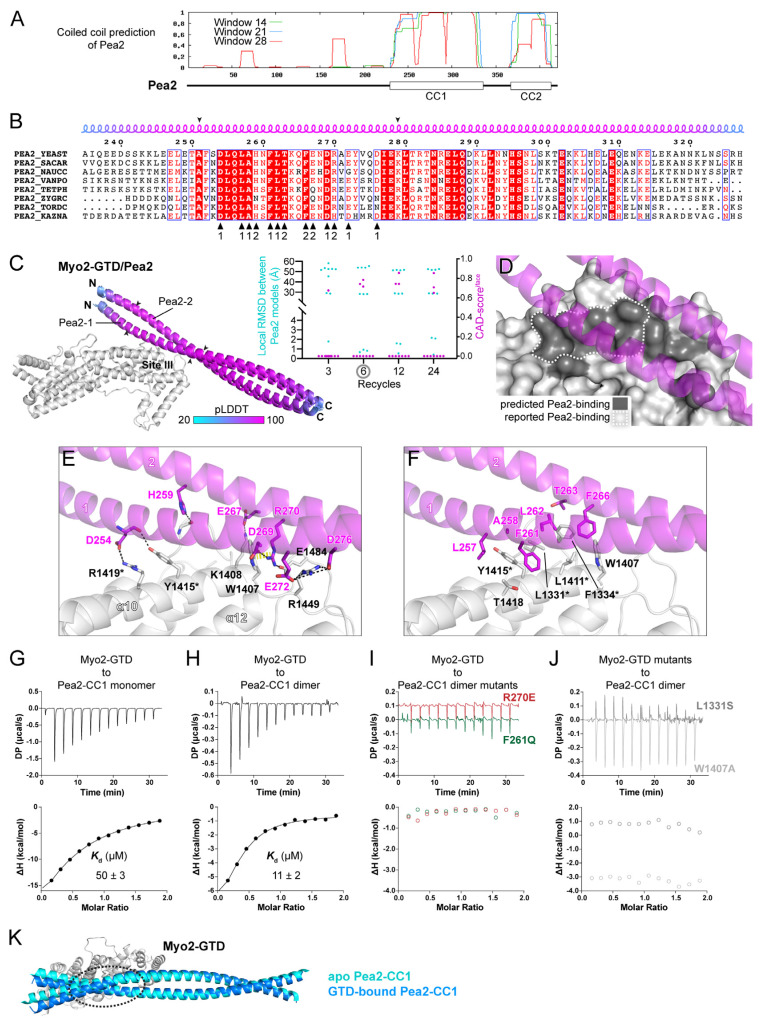
Prediction and analysis of the Myo2-GTD and Pea2-CC1 complex. (**A**) The sequence analysis of Pea2 predicting two coiled-coils (CC1 and CC2) in the C-terminal region. (**B**) Sequence alignment of the Pea2-CC1 among different species. The residues from two protomers that are involved in the binding of the Pea2-CC1 dimer to Myo2-GTD in the complex structure are indicated by black triangles. The boundary used for local RMSD calculation is indicated by two arrowheads. (**C**) The three predicted structures of the Myo2-GTD and dimeric Pea2-CC1 complex. The structures were superimposed by aligning their Myo2-GTD structures and the Pea2-CC1 dimer is colored according to pLDDT value of residues. Two protomers of Pea2-CC1 dimer are named Pea2-1 and Pea2-2, respectively. The CAD-scores^iface^ and the local RMSD of the GTD-bound boundary of Pea2-CC1 were calculated between each of the two predicted models. (**D**) The Pea2-binding surface on Myo2-GTD. The predicted Pea2-binding surface covers the reported one. (**E**,**F**) Molecular details of the Myo2-GTD/Pea2-CC1 interface in the predicted structure, including the polar (**E**) and hydrophobic (**F**) interactions. The two protomers of Pea2-1 and Pea2-2 are labeled as 1 and 2, respectively, and the reported interface residues are marked with asterisks. (**G**,**H**) ITC-based measurements of the binding affinity between monomeric (**G**) or dimeric (**H**) Pea2-CC1 and Myo2-GTD. (**I**,**J**) The ITC-based analysis of the disruptive binding between Myo2-GTD and mutated Pea2-CC1 dimer (**I**) or between mutated Myo2-GTD and Pea2-CC dimer (**J**). (**K**) Structural comparison of the predicted structures of apo- and GTD-bound Pea2-CC1. The GTD-bound site of Pea2-CC1, which is indicated by a dashed circle, shows a substantial change upon GTD binding, while the regions that are not involved in GTD binding have little structural differences.

**Figure 5 biomolecules-12-01032-f005:**
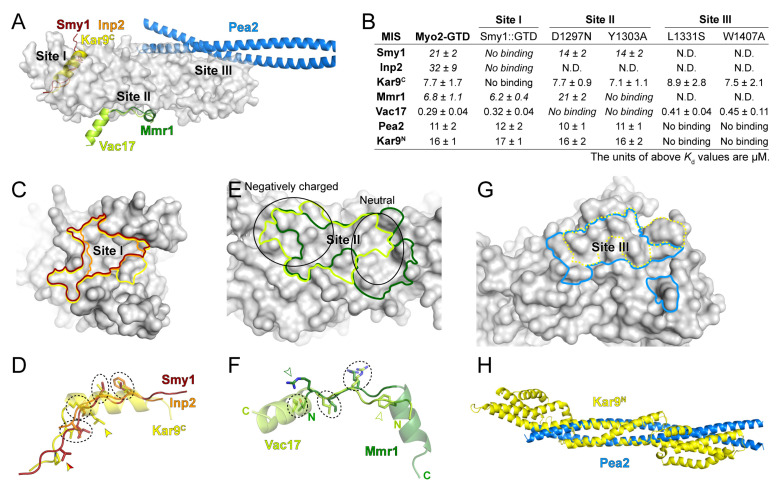
The diverse cargo-binding modes of Myo2-GTD. (**A**) Structural superimposition of six indicated cargo adaptors in complex with Myo2-GTD, showing the three cargo-binding sites of Myo2-GTD. (**B**) Summary of the binding affinities between the indicated cargo adaptors and the Myo2-GTD variants, all of which were detected by ITC-based measurements. (**C**,**D**) Comparison of the cargo-binding surfaces on Myo2-GTD (**C**) and the interacting residues (**D**) of Smy1, Inp2 and Kar9^C^ at site I. The cargo-binding surfaces on Myo2-GTD were highlighted with the same color code used in **A**. The interacting residues that are located at the similar positions in the adaptors are circled by dash lines while the specific interacting residues are labeled by arrowheads. (**E**,**F**) Comparison of the cargo-binding surfaces on Myo2-GTD (**E**) and the interacting residues (**F**) of Mmr1 and Vac17 at site II. The surface properties at different parts of site II are indicated by two black circles in (**E**). (**G**) Comparison of the Pea2 dimer and reported Kar9 binding surface of Myo2-GTD at site III. The latter one was outlined by a dot line. (**H**) Structural comparison of the Pea2-CC1 dimer and Kar9^N^ (PDB ID: 7AG9).

## Data Availability

All data are contained in the manuscript and Appendix A.

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
