# Peer review of "Cargo Recognition Mechanisms of Yeast Myo2 Revealed by AlphaFold2-Powered Protein Complex Prediction"

_biomolecules, 2022, doi:10.3390/biom12081032_

Round 1

Reviewer 1 Report

This manuscript investigated the structures of the S. cerevisiae Myo2-GTD/cargo adapter complexes using AlphaFold2-based prediction. With the strategy, the authors were able to show the previously unknown binding modes of Vac17, Kar9 and Pea2 to Myo2-GTD. The predicted structures were supported by their biochemical analyses including binding affinity measurements. Further, they demonstrated various cargo recognition mechanisms of Myo2, by comparing the Myo2-GTD structures in complex with different cargo adapters.

Overall, the manuscript is clearly written, and their main conclusion is supported by data. This study will provide an efficient tool to understand molecular basis of cargo recognition mechanisms of yeast Myo2. I have a couple of minor comments shown below: 

-       Axis labels in ITC-based measurements graphs (Fig. 2G-H, Fig.3G-I, Fig.4G-J) are hard to read.

-       The table (or the caption) in Fig. 5 (5B) is missing units (µM) for Kd.   

Author Response

  1. Axis labels in ITC-based measurements graphs (Fig. 2G-H, Fig.3G-I, Fig.4G-J) are hard to read.

Response: We have enlarged the labels in the mentioned ITC graphs.

  1. The table (or the caption) in Fig. 5 (5B) is missing units (µM) for Kd.   

Response: We thank the reviewer for pointing out this error. We have stated the unit information by adding a footnote in Figure 5B.

Reviewer 2 Report

The study by Liu et al. reports the cargo recognition mechanisms of yeast myo2 by using AlphaFold2 prediction. The authors performed benchmarking of the known complexes for myo2 before the prediction of the cargo complexes using AlphaFold2. Importantly, the authors predicted the structures of Myo2-GTD in complex with three cargo adaptors. Predictions were further evaluated by biochemical experimental approaches. The study is well designed and clearly written and I have only minor comments.

Comments:

1.    All ITC experimental data were fitted with a one-site binding model whereas the complex structure predictions for Myo2-GTD and Pea2-CC1 dimer were done with a binding ratio of 1:2. The authors should comment on the rationale for this discrepancy in the manuscript.

2.    The authors should provide the database IDs of the targets and the boundaries of the GTD used for complex predictions. Please also include PDB IDs wherever necessary.

3.    The authors should discuss their complex predictions in terms of the myosin enzymatic cycle.

4.    The authors should deposit the script files generated by AlphaFold2 for each step of their predictions as an appendix or supplementary file.

5.    During the predictions, how far apart are the molecules in the complexes placed for the prediction of the complex, especially for the benchmarking? Since the ColabFold uses prior structural information for alignments, how reliable are the RMSDs when compared to the already known crystal structures?

6.    Please include predicted or apparent molecular weight for each chromatogram generated by aSEC in Figure S4A.

7.    The authors should remove the track changes in the supplementary file.  

Author Response

  1. All ITC experimental data were fitted with a one-site binding model whereas the complex structure predictions for Myo2-GTD and Pea2-CC1 dimer were done with a binding ratio of 1:2. The authors should comment on the rationale for this discrepancy in the manuscript.

Response: The two protomers in the Pea2-CC1 dimer form an integrated surface for Myo2-GTD binding (Figure 4E&F). Therefore, we used the one-site binding model for ITC data fitting. Interestingly, although the Pea2-CC1 dimer should have two identical GTD-binding surfaces, our ITC data showed that the binding ratio of 1:2 rather than 2:2, indicated by the N value of ~0.5 (Figure 4H). We suspected that the binding of GTD on one GTD-binding surface of the Pea2-CC1 dimer may slightly change the Pea2-CC1 conformation, which influences the other GTD-binding surface and inhibits the Pea2-CC1 dimer to bind to the second GTD. Indeed, we found that the conformational change of Pea2-CC1 between the apo form and the GTD-bound form (Figure 4K in the revision). We have added this part in the revised manuscript.

  1.  The authors should provide the database IDs of the targets and the boundaries of the GTD used for complex predictions. Please also include PDB IDs wherever necessary.

Response: As suggested, we have added the UniProt IDs and boundaries information of Myo2 and the cargo adaptors in the Table S1. Also, we have included the related PDB IDs in figure legends.

  1.  The authors should discuss their complex predictions in terms of the myosin enzymatic cycle.

Response: As suggested, we have added a paragraph to mention the potential regulation of the Myo2 motor activity in the Discussion section.

  1.  The authors should deposit the script files generated by AlphaFold2 for each step of their predictions as an appendix or supplementary file.

Response: We have added the predicted models analyzed in Figure Figure1 to 5 and Figure S2 to S6 as the supplementary materials.

  1.  During the predictions, how far apart are the molecules in the complexes placed for the prediction of the complex, especially for the benchmarking? Since the ColabFold uses prior structural information for alignments, how reliable are the RMSDs when compared to the already known crystal structures?

Response: The complex prediction in ColabFold does not need the placement of the molecules in certain positions in prior to the calculation.

Indeed, AlphaFold2 were trained using the prior structural information in the PDB. However, the training was based on single chain rather than complexes (Jumper J, Evans R, Pritzel A, Green T, Figurnov M, Ronneberger O, et al. Highly accurate protein structure prediction with AlphaFold. Nature. 2021;596:583-9.). It may explain why the GTD/Inp2 complex structure cannot be successfully predicted in the benchmarking, presumably because the training used the apo-form structure of Myo2-GTD, which adopts a conformation unsuitable for Inp2 binding (Figure S3C&D). Even in the two successful cases, the predicted models converge to the experimental structures only after sufficient recycles calculation (Figure 1B). Hence, the RMSDs is still useful for us to evaluate the prediction reliability and optimize our prediction strategy during the benchmarking.

  1.   Please include predicted or apparent molecular weight for each chromatogram generated by aSEC in Figure S4A.

Response: As suggested, we determined the experimental molecular weights of Myo2-GTD, Vas17-MIS and the mixture by using multi-angle static light scattering and indicated their theoretical molecular weights in the revised Figure S4A.

  1.   The authors should remove the track changes in the supplementary file.  

Response: We have removed the track changes in the revised supplementary file.

Reviewer 3 Report

The manuscript describes how computational structure modeling and experimental ITC measurements are combined to investigate the interactions between Myo2 and other proteins. I liked the combination of the two approaches, and it is important that structure modeling results are later confirmed by the experimental protein interaction studies.

I would suggest a few small improvements to make the description of the research more comprehensible for the readers, thus also enlarging the impact of the article:

* The resulting structural models of protein-peptide complexes should be made available, possibly by providing them as the supplementary materials of the article. It is not necessary to provide all the generated models, but at least the representative ones for each protein complex, on which the conclusions are based.

* It would be nice to see the UniProt ACs in the table S1. The readers could readily get the sequences of the investigated proteins, if UniProt ACs and corresponding domain boundaries would be provided. It would be also nice to see the PDB IDs of the reported experimental protein structures, not only the article references.

* The description of methods can be improved to ensure the reproducibility of the modeling. For example, it is not clear, which AlphaFold neural network models were used, monomeric or multimeric? MMseqs2 method should be cited. I did not understand, how were the PAE values used? Would it be better to use the pTM or iPTM scores, produced by AlphaFold? I do not know if they are reported by ColabFold, there may be some difficulties in getting them. Also, maybe the PAE values for the interaction interface region could be analyzed explicitely, as it is done by computing the ipTM scores in AlphaFold-Multimer? This could help in making decisions about the correctness of the protein-peptide interaction model.

* To benchmark the ability of AlphaFold protein-peptide interaction predictions using experimental structures is a good idea. Importantly, the utilized experimental structures were not included in the training set of AlphaFold, as they appeared in 2019. Nevertheless, two of three tested cases were successfully predicted. It would be interesting to see a more detailed analysis, what went well and what failed in the case of Inp2. For example, to evaluate models of lower quality using only RMSD values is not a very good idea. In such cases methods that compare inter-chain contacts would be probably better suitable. As the calculation of inter-chain LDDT values might be hard, I would suggest to use a similar method, CAD-score (https://bioinformatics.lt/cad-score/), to evaluate the quality of the models. CAD-score allows simple comparison of interaction interfaces, given that the residue numbering is the same in the structures. It can also capture if the binding site is predicted correctly (as described in https://dx.doi.org/10.1007/978-1-0716-0270-6_6).

* I found the summarizing section (3.5) very interesting. I identified the three binding sites while analyzing the interactions of Myo2-GTD in the PPI3D web server (https://bioinformatics.lt/ppi3d/site/detailed/single_sequence/FzH5MYhu/1/null/1/1/1/1). I see that different partners bind to these binding sites. It would be interesting to read a short discussion if the biological mechanisms are different when the protein binds to Site 1, Site 2 or Site 3, but maybe this is out of the scope of this publication. Analogously, it would be interesting to know if the peptide sequences predicted to interact with the same binding sites on Myo2-GTD molecule are similar.

Some text improvements are necessary:

* Line 73: "Recently, AlphaFold2 is emerging as a powerful tool in protein structure prediction with a high accuracy [21]. Several advanced applications are subsequently developed to predict the structure of protein complex, including RoseTTAFold [22], ColabFold [23], and AlphaFold-Multimer [24]."

Something seems wrong in the sentence beginning with "Recently, AlphaFold2 is emerging...". Also, only AlphaFold-Multimer is a specialized application to predict structures of protein complexes. RoseTTAFold is a method for structure prediction of monomeric proteins, and ColabFold is just another method to run AlphaFold2 or AlphaFold-Multimer.

* Line 98: "genetic databases"
This is a bit of AlphaFold jargon, I would use the term "protein sequence databases". I would be also good to specify, which databases were used to build a multiple sequence alignment.

* Line 100: "PEA value"

AlphaFold produces PAE values, not PEA.

To conclude, I liked the article and I think that it can be published after these small yet very significant improvements.

Author Response

  1. The resulting structural models of protein-peptide complexes should be made available, possibly by providing them as the supplementary materials of the article. It is not necessary to provide all the generated models, but at least the representative ones for each protein complex, on which the conclusions are based.

Response: We have added the predicted models analyzed in Figure Figure1 to 5 and Figure S2 to S6 as the supplementary materials.

  1. It would be nice to see the UniProt ACs in the table S1. The readers could readily get the sequences of the investigated proteins, if UniProt ACs and corresponding domain boundaries would be provided. It would be also nice to see the PDB IDs of the reported experimental protein structures, not only the article references.

Response: We have supplemented the UniProt IDs of our target proteins in Table S1. The related PDB IDs were included in figure legends.

  1. The description of methods can be improved to ensure the reproducibility of the modeling. For example, it is not clear, which AlphaFold neural network models were used, monomeric or multimeric? MMseqs2 method should be cited. I did not understand, how were the PAE values used? Would it be better to use the pTM or iPTM scores, produced by AlphaFold? I do not know if they are reported by ColabFold, there may be some difficulties in getting them. Also, maybe the PAE values for the interaction interface region could be analyzed explicitly, as it is done by computing the ipTM scores in AlphaFold-Multimer? This could help in making decisions about the correctness of the protein-peptide interaction model.

Response: As suggested, we have added the information of monomeric neural network model in the Methods and Materials and cited the MMseqs2 method.

PAE value is one of the common AlphaFold2 confidence measures, which shows the confidence level (blue to red indicates confident to unconfident) of relative position of two regions in protein or protein complex, thus implying the potential interaction of these two regions. The plot of PAE value has been used to help judging the prediction quality of a complex (Mirdita M, Schütze K, Moriwaki Y, Heo L, Ovchinnikov S, Steinegger M. ColabFold: making protein folding accessible to all. Nat Methods. 2022 Jun;19(6):679-682.). Since our main purpose is to find the potential MIS region in cargo adaptor, we chose the PAE value to initially evaluate the prediction. To help reader to understand our purpose of using the PAE value, we have included our explanation in the revised manuscript.

The pTM score is used to indicate the prediction confidence and rank the models. Compared with the global evaluation of using pTM, the pLDDT score indicates the local prediction reliability. As we focus on the MIS region in the complex prediction, we selected pLDDT rather than pTM to show the prediction confidence. As for ipTM, because we used ColabFold for all predictions in our study, the ipTM score is not applicable for us to analyze. In addition, to further evaluate our prediction reliability, we follow the reviewer’s suggestion to calculate CAD-scores of all predicted models.

  1. To benchmark the ability of AlphaFold protein-peptide interaction predictions using experimental structures is a good idea. Importantly, the utilized experimental structures were not included in the training set of AlphaFold, as they appeared in 2019. Nevertheless, two of three tested cases were successfully predicted. It would be interesting to see a more detailed analysis, what went well and what failed in the case of Inp2. For example, to evaluate models of lower quality using only RMSD values is not a very good idea. In such cases methods that compare inter-chain contacts would be probably better suitable. As the calculation of inter-chain LDDT values might be hard, I would suggest to use a similar method, CAD-score (https://bioinformatics.lt/cad-score/), to evaluate the quality of the models. CAD-score allows simple comparison of interaction interfaces, given that the residue numbering is the same in the structures. It can also capture if the binding site is predicted correctly (as described in https://dx.doi.org/10.1007/978-1-0716-0270-6_6).

Response: We thank the reviewer for the wonderful suggestion of using CAD-score. We calculated the CAD-scores of all predicted models, which further confirms our conclusion based on the RMSD calculation. We have added the CAD-scores in Figure 1-4 of our revision. 

  1. I found the summarizing section (3.5) very interesting. I identified the three binding sites while analyzing the interactions of Myo2-GTD in the PPI3D web server (https://bioinformatics.lt/ppi3d/site/detailed/single_sequence/FzH5MYhu/1/null/1/1/1/1). I see that different partners bind to these binding sites. It would be interesting to read a short discussion if the biological mechanisms are different when the protein binds to Site 1, Site 2 or Site 3, but maybe this is out of the scope of this publication. Analogously, it would be interesting to know if the peptide sequences predicted to interact with the same binding sites on Myo2-GTD molecule are similar.

Response: To understand the biological mechanisms of different cargo binding sites is indeed a motivation of this study. Our structural comparison showed that the site III is closed to the interface of head/tail interaction in Myo2, which inhibits the motor activity of Myo2 (Donovan KW, Bretscher A. Head-to-tail regulation is critical for the in vivo function of myosin V. J Cell Biol. 2015 May 11;209(3):359-65). Thus, the cargo adaptors that bind to the site III potentially induce the activation of Myo2. We have added this information in the Discussion section.

As our previous paper (Tang K, Li YJ, Yu C, Wei ZY. Structural mechanism for versatile cargo recognition by the yeast class V myosin Myo2. J Biol Chem. 2019;294:5896-906.) have already reported that the sequences of MISs that bind to the site I show certain similarity, we did not further discuss the point in this study. As for the site II, Vac17 and Mmr1 have different binding modes (Figure 5F) and share little sequence similarity.

Some text improvements are necessary:

  1. Line 73: "Recently, AlphaFold2 is emerging as a powerful tool in protein structure prediction with a high accuracy [21]. Several advanced applications are subsequently developed to predict the structure of protein complex, including RoseTTAFold [22], ColabFold [23], and AlphaFold-Multimer [24]."

Something seems wrong in the sentence beginning with "Recently, AlphaFold2 is emerging...". Also, only AlphaFold-Multimer is a specialized application to predict structures of protein complexes. RoseTTAFold is a method for structure prediction of monomeric proteins, and ColabFold is just another method to run AlphaFold2 or AlphaFold-Multimer.

Response: We are sorry for the misleading description. AlphaFold2 was initially developed for single chain protein prediction. Independently, RoseTTAFold was developed for both single chain protein and protein complex predictions. AlphaFold-Multimer is a refined version of AlphaFold2 for the prediction of protein complexes and ColabFold is an easy-to-use application to predict protein complex structures by combining the fast homology search of MMseqs2 with AlphaFold2 or RoseTTAFold (Mirdita M, Schütze K, Moriwaki Y, Heo L, Ovchinnikov S, Steinegger M. ColabFold: making protein folding accessible to all. Nat Methods. 2022 Jun;19(6):679-682.). We have modified this paragraph as “Recently, AlphaFold2 and RoseTTAFold are emerging as powerful tools in protein structure prediction with a high accuracy. Based on them, the applications including ColabFold and AlphaFold-Multimer are subsequently developed to predict structures of protein complexes.”.

  1. Line 98: "genetic databases"

This is a bit of AlphaFold jargon, I would use the term "protein sequence databases". I would be also good to specify, which databases were used to build a multiple sequence alignment.

Response: We have changed the term with “protein sequence databases including UniRef90” in the revised manuscript to indicate the specific databases used in ColabFold.

  1. Line 100: "PEA value"

AlphaFold produces PAE values, not PEA.

Response: We have corrected the typo.

Round 2

Reviewer 3 Report

The authors responded to all of my questions and comments.

Interestingly, the monomeric version of AlphaFold was used to predict the structures of protein-peptide complexes. The resulting models were accurate enough, but it would be interesting to see if using AlphaFold-Multimer, which is dedicated for modeling of protein complexes, would improve the models in the cases where the results were unsatisfactory. I think that newer versions of ColabFold can use AlphaFold-Multimer, but probably this is just a suggestion for future investigations.